# Zika Vaccine Development—Current Progress and Challenges for the Future

**DOI:** 10.3390/tropicalmed4030104

**Published:** 2019-07-14

**Authors:** Joel N. Maslow

**Affiliations:** 1GeneOne Life Science, Inc., Seoul 06060, Korea; jmaslow@geneonels-us.com; Tel.: +1-484-965-9147; 2Department of Medicine, Morristown Medical Center, Morristown, NJ 07960, USA

**Keywords:** Zika virus, flavivirus, emerging infectious disease, vaccine, sample size

## Abstract

Zika virus is an emergent pathogen that gained significant importance during the epidemic in South and Central America as unusual and alarming complications of infection were recognized. Although initially considered a self-limited benign infection, a panoply of neurologic complications were recognized including a Guillain–Barré-like syndrome and in-utero fetal infection causing microcephaly, blindness, and other congenital neurologic complications. Numerous Zika virus vaccines were developed, with nine different vaccines representing five different platforms entered into clinical trials, one progressing to Phase II. Here we review the current landscape and challenges confronting Zika virus vaccine development.

## 1. Introduction

The Zika virus, discovered in Uganda in 1947 [1], was shown to be endemic through Sub-Saharan Africa and tropical areas of Southeastern Asia in studies through the second half of the 20th century [2]. Isolated outbreaks occurred in Yap Island in 2007 [3] and on French Polynesia in 2014 [4]. Starting in mid-2015, Zika virus infection achieved epidemic status, spreading rapidly through South America, Central America, and the Caribbean Islands [5]. It was soon recognized that Zika virus infection occurring during pregnancy caused microcephaly and other congenital disorders in the developing fetus, the latter being the primary reason for the World Health Organization (WHO) labelling Zika as an international threat in early 2016. Beginning in late 2015, numerous academic labs and pharmaceutical companies initiated work to develop a vaccine against Zika, however, by the time the first vaccines entered clinical trials, the Zika epidemic had started to wane creating significant challenges to vaccine assessment [6] that has engendered discussion of other regulatory pathways to licensure [7].

In this paper, we provide a brief update of current progress in Zika virus vaccine development and explore the challenges to vaccine assessment and eventual licensure.

## 2. Zika Vaccine Target Population

Zika virus infection presents with a symptom complex consisting of a diffuse maculo-papular rash, fever, asthenia, myalgias, arthralgias, headache, and retroorbital pain. The frequency and degree of symptomatology has, however, varied between studies. A retrospective study immediately following the outbreak on Yap Island found that only 19% of survey participants reported symptoms consistent with Zika virus infection [3], a figure that has been cited to suggest that Zika virus infection is asymptomatic in as many as 80% of individuals. However, of 557 individuals who provided blood samples, representing 16% of households, 38% were symptomatic [3]. A second retrospective study of the larger outbreak in French Polynesia [8,9], similarly appeared to have a low rate of symptomatology based on a sample of blood donors [10]. However, a subsequent seroprevalence study found that 43% of those with evidence of prior infection had symptoms consistent with Zika virus infection [11]. A more recent meta-analysis of 23 studies noted that between 0 and 83% of cases were reported as asymptomatic, although as the authors note, assessment of the prevalence of symptoms was not the goal of many studies [12].

In adults, the most common reported complication of Zika virus infection is a Guillain–Barré-like illness that had a prevalence of 0.24 cases per 1000 cases of infection in the French Polynesian epidemic [13]. Of interest, a recent meta-analysis has questioned this causal association [14]. Other less common reported neurologic complications in adults include meningoencephalitis [15] and an ADEM illness.

In contrast, infection during pregnancy has been associated with fetal microcephaly and a number of other congenital illnesses including visual deficits, hearing disorders, neural calcifications, learning disabilities, and arthrogryposis that may affect as many as 30% children born to mothers infected during pregnancy [16,17,18,19,20,21]. Additionally, Zika can directly invade the placenta and has been associated with prolonged maternal viremia [22,23,24].

Zika virus can also be transmitted through sexual contact, first reported for a researcher returning from Senegal in 2008 [25]. Numerous subsequent case reports were published among travelers as part of the epidemic affecting the Americas [26]. The frequency of sexual transmission in endemic regions is unknown and could not be differentiated from mosquito-borne transmission. Zika virus carriage in the male urogenital tract is common through 90 days post infection and may persist in some to 9 months [27].

In mice, Zika virus infection causes testicular atrophy and significantly decreases spermatic function and fertility rates [28,29,30]. In one study of a Zika virus DNA vaccine, the adverse effects in the male reproductive system were prevented by vaccination [30]. The question of whether Zika virus can adversely affect human reproductive potential and, if so, whether such effects would be age-related is unknown.

Females may also excrete Zika virus RNA for extended times. A prospective study of five women showed that Zika RNA was detected in vaginal fluid for as long as 6 months and a month or more in three of the five [31]. Murine studies showed that Zika virus caused infection of the ovaries of non-immunosuppressed C57Bl/6 mice and induced a T-cell inflammatory reaction, but without affecting reproductive potential [32]. In contrast to the above human study, female macaques rapidly cleared virus from the genital tract [33].

Thus, while Zika virus infection is mildly symptomatic and self-limited for the vast majority of individuals, with infrequent neurologic complications in adults, vaccination of the general population may not be warranted. However, vaccination of females at or entering reproductive age and their male partners is prudent. As discussed previously, vaccination during pregnancy is non-ideal due to the time to generate protective immunity and unknown vaccine safety [34]. The one unknown aspect is whether vaccination of males of any age may be beneficial to protect against testicular complications. Calls for generalized vaccination programs have, however, been put on hold due to the decreasing incidence of Zika infection rates, as discussed in the next section.

## 3. Changing Epidemiology of the Zika Virus Outbreak

In late 2015 and through mid-2016, global concern for this spreading epidemic was increasing, however, the experience in the two prior epidemics was predictive of the subsequent epidemiology in the Americas. The Zika virus outbreaks in Yap Island and French Polynesia were characterized by rapid onset and rapid resolution over a span of 3–4 months [3,9]. The Zika epidemic in the Americas was characterized by a longer time to reach peak incidence, taking approximately one year for Brazil [35] presumably related to the larger and more varied geography, with attack rates decreasing greater than 100-fold the next year [35]. Data published by the Pan American Health Organization (PAHO) found similar patterns through South and Central America (https://www.paho.org/). The almost complete disappearance of Zika has created significant barriers to ongoing vaccine studies as attack rates have fallen below those need to meet reasonable sample sizes [6].

The WHO removed the designation of Zika a virus of global concern in 2018 as case rates have fallen. In March 2018, the WHO and the National Institutes of Allergy and Infectious Diseases (NIAID) convened a meeting of academics, representatives from industry, and regulators to discuss the approach to Zika virus vaccine development in an era of waning incidence [7]. Although Zika virus remains endemic in Asia, Africa, and Central and South America, current transmission rates, coupled with high background herd immunity, would require an extremely large and logistically difficult study. As population immunity wanes, future outbreaks are likely. The ability to conduct a meaningful future clinical trial may be dependent upon having pre-existing site and regulatory approvals with an established vaccine network to enable rapid response to new reports suggesting increased disease activity.

## 4. Zika Vaccine Candidates in Development

Vaccine development against the Zika virus began in earnest in late 2015 following the reports of microcephaly in fetuses and infants in Brazil. Of note, the first demonstration of immunoprotection was as part of a 1953 study to define the ultrastructural characteristics of Zika virus, that found intramuscular vaccination of mice with infectious viral filtrates protected against cerebral infection [36].

Poland et al. provide a comprehensive listing of almost 40 Zika virus vaccines in development as of 2017 [37]. Diamond et al. discuss potential immunoreactive epitopes on the Zika envelope and provide further information on those vaccines that progressed into clinical trials [38]. The large number of delivery platforms include live attenuated and inactivated whole-viruses; viral-vectored vaccines utilizing adeno-associated virus, vesicular stomatitis virus, measles virus, and dengue or Yellow Fever chimeric vaccines; DNA and mRNA vaccines; and peptide and protein subunit vaccines. Most have been assessed in animal models utilizing non-human primates and/or lethal challenge experiments involving immunosuppressed mice [37]. As of 2017, six vaccines had advanced to Phase I studies [37,39,40]. Since that time, two additional vaccines have entered into clinical trials (Table 1).

A total of three DNA vaccines have entered into human testing [41,42], including one that has advanced to Phase II (Table 1).

GLS-5700, a DNA vaccine encoding for the Zika virus prME genes designed as a consensus based on available Zika virus sequences through December 2015 [43], was the first to enter into clinical trials. In pre-clinical studies, vaccinated mice and non-human primates were shown to develop B and T-cell immune responses against the Zika virus envelope and protected against development of neurologic disease and death in immunosuppressed, interferon α, β receptor deficient (IFNAR) mice [43]. Moreover, histologic sections of brain tissue showed that vaccinated mice were without inflammatory infiltrates evident in non-immunized mice [43]. Subsequent studies showed that the vaccine protected against testicular damage, testicular atrophy, spermatozoal damage, and infertility in mice [28,30]. A Phase I study evaluated GLS-5700 administered via intradermal injection (ID), followed by electroporation (EP) at doses of either 1 or 2 mg per vaccinations followed immediately by electroporation at baseline, 4 weeks and 12 weeks [42]. There were no serious adverse events (SAE) reported as part of the study, with the most frequent adverse events related to discomfort, pain, or swelling at the injection site. Seroconversion was observed in 83% of individuals after two vaccinations and 100% after three vaccinations with GLS-5700 [42]. Neutralizing antibodies were detected for 62% of participants in a Vero-cell (monkey kidney cell) assay, however, 95% of participants demonstrated the ability to neutralize infection of U87MG neuroblastoma cells [42]. Vaccine responses were maintained through a year of follow-up. There was no difference in responses based on dose level. Notably, passive transfer of immune serum from vaccinated participants was able to protect 92% of IFNAR mice against lethal infection independent of the presence of Vero cell neutralizing antibodies [42]. GLS-5700 is being evaluated as part of a second double-blind, placebo-controlled clinical trial (NCT02887482) performed in Puerto Rico. Analysis of the latter study is in progress.

Two additional DNA vaccines, based on the French Polynesian H/PF/2013 strain were developed as chimeras that included the JEV prM signal sequence followed by the Zika envelope (E) gene (VRC5283) or a similar construct with the terminal 98 amino acids of E, representing the stem and transmembrane regions, exchanged for the analogous JEV sequence (VRC5288) [44]. Both vaccines were immunogenic for mice and NHPs and protected >90% of NHPs against viremia at a dose of either 1 or 4 mg given twice [44]. Both vaccine candidates were advanced into clinical trials with 4 mg of administered intramuscularly at weeks 0, 4 and 8 with vaccine VRC5288 administered by needle and syringe while VRC5283 was administered either as a single dose or split dose by needle and syringe or as a split dose given by the Pharmajet needle-free device [41]. Seroconversion was 100% in the group administered vaccine with the needle-free device, less for vaccine administered as split-dose by needle and syringe, and lowest for vaccine given as a single injection with needle and syringe. The VRC5283 vaccine was advanced into Phase II studies in the Americas that utilized 140 clinical sites with clinical site selection guided by epidemiologic modeling [38]. Long-term follow-up has not been reported.

Three inactivated vaccines have entered into clinical studies, of which clinical data for only ZPIV vaccine has been published [45]. ZPIV is a whole inactivated virus vaccine of Puerto Rican strain PRVABC59 [46]. Studies in mice showed that a single vaccination given intramuscularly with alum generated antibody titers of approximately 3.7 log10 and fully protected Balb/C immunocompetent mice from viremia, whereas unvaccinated mice were unprotected and subcutaneously vaccinated mice were only partially protected against the Zika Brazil strain [46]. A subsequent study in non-human primates vaccinated twice at four-week intervals with alum generated binding and microneutralization antibody titers of 3.54 and 3.55 log10, respectively, and complete protection against viremia and viruria following challenge with either Brazilian or Puerto Rican strains of Zika virus [47]. ZPIV safety and immunogenicity was tested in three clinical trials to assess Zika vaccine responses relative to dose level, vaccination schedule, or following vaccination with either the Yellow Fever YF-VAX or Japanese encephalitis virus (JEV) IXARO vaccines to assess responses in a flavivirus-exposed population [45]. There were no vaccine-associated SAEs reported. The most common adverse events were pain and tenderness at the injection site; no neurologic events were reported. Seroconversion was 92% using a cutoff for peak geometric mean titer of 1:10 and 77% using a titer of 1:100. Response rates after a single immunization were 11% and 5.5% using cutoffs of 1:10 or 1:100, respectively. Vaccine responses were observed through day 57. Passive transfer of purified IgG derived from 10 Zika immunized participants to groups of 5 Balb/C mice per participant provided sterilizing immunity against viremia for 82% (41 of 50) of mice overall, with viremia observed for one or more mice per group inoculated with sera from five (50%) individuals [45].

Clinical trial data for the other vaccines has not yet been reported as of the date of this monograph, however, pre-clinical data has been reported for three candidate vaccines. An mRNA vaccine that incorporates the prM-E genes of a Micronesian strain of Zika virus was created incorporating with or without four-point mutations in the fusion-loop segment of the DII region of the envelope gene that abolished binding of antibodies directed against the fusion-loop region to reduce the potential risk for antibody-dependent enhancement of infection [48]. Immunization of AG129 mice with un-modified lipid nanoparticle-encapsulated vaccines mRNA was immunogenic and protective against lethal infection; immunization of C57BL/6 immunocompetent mice followed by treatment with anti-ifnar1 blocking antibody showed protection against viremia in approximately 60% of animals [48]. PIZV, an inactivated vaccine derived from Puerto Rican strain PRVABC59 selected as without passage-related mutations, protected against lethal Zika virus challenge in AG129 mice when administered with alum adjuvant [49]. A measles-vectored vaccine encoding the Zika virus prME was shown to lessen viremia in pregnant IFNAR mice and prevented clinical disease in mouse pups [50]. Preclinical data for VLA-1601 inactivated viral vaccines and the rZIKV/D4Δ30-713 live virus vaccine has not yet been reported.

## 5. Potential Animal Models for Vaccine Evaluation

A number of animal models have been assessed for vaccine development and have been reviewed elsewhere [51]. Immunocompetent mice can develop transient viremia following infection and may represent models to test sterilizing immunity. Interferon α/β receptor knockout mice (AG129 or IFNAR -/-) develop lethal infection following Zika virus infection and have been used as a more stringent measure of protection. Non-human primates develop a self-limited mild illness associated with viremia and can transmit virus in utero to primate fetus [52].

## 6. Challenge Human Infection Models

The logistical difficulties in pursuing standard vaccine evaluation have created significant interest in the possibility of controlled human infection models (CHIM). The conduct and planning for human challenge trials raises unique medical and ethical considerations. For Zika virus trials, one must balance the relative benign and self-limited infection experienced by the vast majority against the risk for less benign complications and transmission risks.

As reviewed above, published studies provide estimates that Zika virus infection is relatively asymptomatic and self-limited for the majority of individuals, however, methodology in these studies varies widely. In adults, two sets of complications warrant consideration for a proposed CHIM study. As reviewed above, a Guillain–Barre-like syndrome occurs in approximately 1 of 5000 Zika virus infections [13], whereas other neurologic complications such as meningoencephalitis, myelitis [15,53,54], and acute disseminated encephalomyelitis [54] are rare. Second, and perhaps more important, is the risk of transmission to a sexual partner and the potential for infection during pregnancy. Zika virus commonly persists in the male urogenital tract for 3 months, and may persist in some individuals for up to 8 months [27]. Some have considered limiting studies to non-pregnant females as Zika virus colonization of the female genital tract may be temporally limited. The fact that Zika virus is known to cause testicular atrophy in mice [28,29,30], raises yet another as yet theoretical concern for humans. These questions as well as the theoretical potential for vector-borne transmission were debated in detail in late 2016 with the conclusion that the benefits of a human challenge infection did not outweigh the risks [55], however, this analysis was performed just as the initial epidemic wave in the Americas was ending. The group published a follow-up in 2018 [56]. Despite the recognition that conducting a placebo-controlled vaccine trial had become significantly more difficult due to declining case rates, the group’s conclusion was essentially unchanged.

To address safety concerns, there has been work to develop attenuated viral strains deleted for potential neurotropic regions [57] with a goal to prevent viremia. Whether the attenuated viral strain (rZIKV/D4Δ30-713) being tested in a Phase I study will serve as putative challenge strain is as yet undetermined.

## 7. Summary

In summary, Zika vaccine development continues with multiple candidate vaccines in clinical trials. Because of the significant decline in incidence, evaluation of vaccine efficacy is increasingly difficult. There has been renewed interest in animal model and human infection models of infection.

## Figures and Tables

**Table 1 tropicalmed-04-00104-t001:** Zika vaccines in clinical development.

Vaccine Type/Vaccine Name	Developer	Development Phase	Clinical Trials Gov Designation
Inactivated virus			
ZPIV	WRAIR	Phase I	NCT02937233
			NCT02952833
			NCT03008122
			NCT02963909
PIZV	Takeda	Phase I	NCT03343626
VLA-1601	Valneva	Phase I	NCT03425149
Attenuated live virus			
rZIKV/D4Δ30-713	NIAID	Phase I	NCT03611946
DNA			
GLS-5700	GeneOne/Inovio	Phase I	NCT02809443
			NCT02887482
ZKADNA085-00-VP	VRC	Phase I	NCT02840487
ZKADNA085-00-VP	VRC	Phase II	NCT02996461
			NCT03110770
mRNA			
mRNA-1325	Moderna	Phase I	NCT03014089
Viral vectored			
MV-Zika (Measles vectored)	Themis	Phase I	NCT02996890

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
