# Peer review of "Zika Vaccine Development—Current Progress and Challenges for the Future"

_tropicalmed, 2019, doi:10.3390/tropicalmed4030104_

Round 1

Reviewer 1 Report

This communication provides a succinct review of the current status of vaccine development for the Zika virus. It begins with a rather cursory summary of the Zika virus pandemic which began in Brazil, eventually spreading throughout many continents. The review also provides a brief description of the congenital Zika syndrome (although not using that term), as well as sexual transmission and effects on the human and experimental animal reproductive system. The manuscript is well-organized and written, and summarizes the major highlights of ZIKV vaccine development and clinical trials.

The references appear current up to 2018, but I identified only 2 references (#23, #42) from 2019... perhaps the authors can supplement their manuscript with some additional 2019 references - Michael Diamond's January 2019 article in the Annual Review of Medicine would be a candidate for one additional reference...

The author briefly hints at the difficulties of performing clinical trials with the decreased global prevalence of ZIka infection... can the authors delve into this issue in greater detail including how to test for vaccine efficacy in the field in the setting of low prevalence of vector-borne infection, and discuss where (if anywhere) Zika prevalence is sufficiently prevalent that these clinical trials can be performed.

The author may also wish to consider the follow comments and suggestions:

Abstract: I would not consider Zika virus infection to be a re-emergent infection... I believe that the literature generally considers it to be an emerging infection.

Introduction: Although initially recognized as occurring in S. America (Brazil) and then spreading throughout Central America and the Caribbean, it was also recognized in other countries and continents, becoming a pandemic. Please revise this to accurately reflect the geographic distribution of infection during this time period.

Author Response

Additional recent references have been added to the review including: 

Diamond 2019 Ann Rev Med, ZIka vaccine review

Reyes 2019 EID, discussing genital carriage in female patients

Caine 2019 JID, discussing ovarian infection in female mice

Bautista 2019 J Neurol Sci, of a meta-analysis of Zika and GBS

Vannice 2019 Vaccine, a summary of the WHO-NIAID Zika conf

The discussion regarding the difficulty of performing clinical trials of Zika vaccines was expanded. 

Abstract: Zika was changed to "an emergent infection". 

Introduction: the historical origins of Zika in Africa and tropical Asia and was added to the 1st paragraph of the introduction.  

Reviewer 2 Report

Your manuscript is a valuable, current, succinct and informative review of the current situation and challenges that face vaccine development for Zika and other flaviviruses.

Line 158,  PIZV, not PZIV (minor typo) needs correction.

Author Response

The typographical error in line 158 was corrected. 

Reviewer 3 Report

Clear and concise monograph. I have just a couple minor comments:

1) Check the word Zika (throughout the paper), some extra capitals (lines 16, 24 etc..)

2) In section of Zika vaccine candidates in development (starting from the line 101), could you describe first the the DNA vaccines all, then others and modify Table 1 accordingly. It is easier for reader to follow your text if all DNA vacc. parts are kept together and those are the first in Table 1, and then something else...

3) Line 158: Is there a sentence missing? Just PZIV and reference? 

Author Response

The capitalization of "Zika" was corrected and the remaining citations were checked for accuracy, 

The section on vaccine candidates was reorganized by vaccine type. This included changes to the introductory paragraphs to ensure consistency and clarity. 

The reference to PZIV was expanded and the paragraph edited to ensure consistency with information presented.